# Mapping the Role of P-gp in Multidrug Resistance: Insights from Recent Structural Studies

**DOI:** 10.3390/ijms26094179

**Published:** 2025-04-28

**Authors:** Shi Ting Tia, Min Luo, Wenjie Fan

**Affiliations:** 1Department of Biological Sciences, National University of Singapore, Singapore 117558, Singapore; e0968993@u.nus.edu; 2Center for Bioimaging Sciences, Department of Biological Sciences, National University of Singapore, Singapore 117543, Singapore

**Keywords:** multidrug resistance, ABC transporter, cryo-electron microscopy, cancer

## Abstract

P-glycoprotein (P-gp/ABCB1), a key ATP-binding cassette (ABC) transporter, plays a central role in multidrug resistance (MDR), one of the leading causes of chemotherapy failure in cancer treatment. P-gp actively pumps chemotherapeutic agents out of cancer cells, reducing intracellular drug concentration and compromising therapeutic efficacy. Recent advancements in structural biology, particularly cryogenic electron microscopy (cryo-EM), have revealed detailed conformational states of P-gp, providing unprecedented insights into its transport mechanisms. In parallel, studies have identified various P-gp mutants in cancer patients, many of which are linked to altered drug efflux activity and resistance phenotypes. This review systematically examines recent structural studies of P-gp, correlates known patient-derived mutations to their functional consequences, and explores their impact on MDR. We propose plausible mechanisms by which these mutations affect P-gp’s activity based on structural evidence and discuss their implications for chemotherapy resistance. Additionally, we review current approaches for P-gp inhibition, a critical strategy to restore drug sensitivity in resistant cancers, and outline future research directions to combat P-gp-mediated MDR.

## 1. Introduction

Cancer remains one of the most significant global health challenges, with an estimated one in five individuals diagnosed in their lifetime [1]. An estimated 35 million new cancer cases are expected to be diagnosed worldwide in 2050, emphasizing the urgent need for effective treatments. Despite advances in chemotherapy, treatment outcomes are often hindered by the development of multidrug resistance (MDR), a phenomenon where cancer cells acquire the ability to evade multiple chemotherapeutic agents [2,3]. MDR is implicated in 90% of chemotherapy failure during late-stage cancer [4], highlighting its critical role in treatment resistance and poor patient outcomes. One of the key players in MDR is P-glycoprotein (P-gp), a member of the ATP-binding cassette (ABC) transporter family. P-gp functions as an efflux pump, utilizing ATP hydrolysis to actively efflux a broad spectrum of chemotherapeutic agents from cancer cells. By reducing intracellular drug concentrations, P-gp diminishes the efficacy of treatment, ultimately contributing to therapy failure [4,5].

Over the last few decades, numerous studies have uncovered a variety of P-gp mutants in cancer patients, many of which are associated with altered drug resistance phenotypes [6,7,8]. Recent breakthroughs in structural biology, particularly through cryogenic electron microscopy (cryo-EM), have revolutionized our understanding of P-gp. High-resolution structures have revealed critical details of its conformational states of substrate/inhibitor binding and ATP hydrolysis. Gaining a detailed understanding of the working mechanism is essential for developing strategies to overcome MDR.

The structural insights provide a foundation for understanding the mechanistic consequences of patient-derived P-gp mutations. However, while research has investigated the impact of various P-gp mutations on patient survival metrics such as overall survival and progression-free survival, a comprehensive analysis that integrates structural data with functional mechanisms remains limited. This gap hinders our ability to fully understand how P-gp mutations drive MDR and to identify effective strategies to overcome resistance.

In this review, we aim to address the role of P-gp in MDR by presenting a comprehensive overview of P-gp structural studies and patient-derived mutations. We examine how these mutations impact P-gp’s function and propose plausible mechanisms for their role in MDR based on recent structural evidence. Recognizing the importance of targeting P-gp for overcoming chemotherapy resistance, we also review existing approaches for P-gp inhibition and identify future research directions. By bridging structural biology, functional mechanisms, and therapeutic perspectives, we provide a unified understanding of P-gp-mediated MDR and offer insights to guide the development of novel strategies to combat drug resistance in cancer therapy.

## 2. Structure of P-gp

### 2.1. Overview of High-Resolution Structural Studies of P-gp

P-gp, a 170 kDa transmembrane protein, was first discovered in 1976 in drug-resistant Chinese hamster ovary cells [9]. P-gp has been found to be prominently expressed in normal organs such as the kidney, liver, colon, and the brain [10]. As drugs are found to have greater impacts in these organs, the function of P-gp is primarily to reduce the accumulation of substrates to reduce toxicity [11,12]. P-gp interacts with a diverse range of substances, with molecular weights varying from 250 to 4000 Da, including important therapeutic agents like paclitaxel, verapamil, doxorubicin, and vincristine [13,14,15,16,17]. This protein exhibits a strong affinity for globular hydrophobic, lipophilic, and weakly amphipathic compounds, due to its binding site, which is abundant in aromatic and nonpolar amino acids like phenylalanine and valine [15,18,19,20].

Since the discovery of P-gp, its structure has been extensively studied using various methods, including X-ray crystallography, spectroscopy methods, and cryo-EM (Table 1). Among these, cryo-EM has shown the greatest potential for elucidating the overall structure of human P-gp [19,21,22,23,24,25,26].

### 2.2. Domain Organization of P-gp

P-gp is a membrane transporter made up of a single polypeptide consisting of 1280 amino acids. It is structured as a full transporter, comprising two transmembrane domains (TMDs) and two nucleotide-binding domains (NBDs), arranged in the following sequence: TMD1 at the N-terminus, followed by NBD1, TMD2, and NBD2 at the C-terminus (Figure 1A) [22,24,39].

### 2.3. Transmembrane Domains (TMDs)

The TMDs play critical roles in recognizing and transporting drugs [32,40]. Each TMD of P-gp consists of six transmembrane (TM) helixes which are hydrophobic alpha helices [22]. TMD1 spans from TM1 to TM6 while TMD2 encompasses TM7 to TM12 (Figure 1A) [21,41]. These TM helices are connected to each other via extracellular loops on the external side of the cell and intracellular loops in the cytoplasm [42,43]. The TMDs play a critical roles in the transport of drugs, forming the pathway through which substrates cross the lipid bilayer membrane [32,40]. A flexible, highly charged linker of 75 amino acids connects the two halves of P-gp. This linker, positioned between the C-terminus of NBD1 and the N-terminus of TMD2, has been largely unresolved in structural studies due to its high mobility. Despite this, its importance is underscored, with several studies identifying it as crucial for trafficking to cell surface, ATPase activity, and transporter function [44,45]. The linker contains multiple phosphorylation sites, and proteolytic cleavage of this region has been shown to influence the ATPase activity of P-gp [46]. Moreover, the linker region participates in protein–protein interactions, with three specific regions near highly conserved sequences interacting with NBD2, TM3, and TM9 [47]. These interactions help the linker to perform a damper function, absorbing significant conformational motions of P-gp, thereby restricting the rotational freedom of NBD2 and stabilizing the cytoplasmic domains [47]. The linker has also been found to be critical for the recognition of substrates and for preventing the entry of substrates into the drug-binding pocket from the cytoplasmic side [47,48].

Substrates can enter P-gp through the lateral gates, also known as the TMD portal (Figure 1A) [22]. These gates are formed by the paired TM4 and TM6 from TMD1, as well as TM10 and TM12 from TMD. In addition to facilitating substrate entry, these gates play a crucial role in selecting and verifying whether a molecule is a suitable substrate for transport [49].

### 2.4. Nucleotide-Binding Domains (NBDs)

The NBDs are highly hydrophilic domains located on the cytoplasmic side of the membrane and they exhibit greater flexibility than the TMDs [50], with a separation of two NBDs at 30 Å in the inward-facing conformation [38]. Each NBD binds and hydrolyzes ATP and both are essential for the proper function of P-gp [51,52]. Studies also suggest that catalysis alternates between the two ATP-binding sites [53]. The NBDs contain several key motifs: the A-loop, walker A, Q-loop, LSGGQ signature motifs, walker B, D-loop, and the H-switch (Figure 1B) [24,54,55]. The A-loop has highly conserved aromatic amino acid residues that interact with the adenine ring of ATP via π-π stacking. Downstream, the walker A, together with walker B, is critical for ATP binding [53,56]. The Q-loop is essential for the detection of the γ-phosphate of ATP and facilitating interactions between NBDs [57]. It strengthens van der Waals forces with the C motif of the opposite NBD, using a magnesium ion and the γ-phosphate to support ATP binding [58,59]. The LSGGQ signature motif, a key characteristic of all ABC transporters, is crucial for both ATP hydrolysis and communication with the TMDs and directly interacts with the nucleotide in its ATP-bound state [51]. The walker B has a highly conserved glutamate residue, also known as a catalytic carboxylate base, that facilitates the binding of Mg^2+^ and the catalytic base of ATP hydrolysis [57,58]. This glutamate residue also facilitates a nucleophilic attack on ATP via a water molecule [57]. Additionally, the D-loop plays a role in facilitating communication between the catalytic sites [57,60]. The H-switch has a histidine residue that act as a catalytic base facilitating ATP hydrolysis [61].

## 3. Central Drug-Binding Cavity of P-gp

The central drug-binding pocket of P-gp is located within two TMDs (Figure 2A) [21]. Recent studies have revealed that P-gp does not possess a single, fixed drug-binding site. Instead, molecules can bind to multiple regions on the transporter, offering significant flexibility and enabling the recognition of a diverse array of substances [16,62,63,64]. This characteristic, known as poly-specificity, is attributed to the transporter’s adaptable drug-binding pocket, which allows it to accommodate a diverse range of drugs.

Recent structural studies have shed light on the detailed drug-binding mechanisms of P-gp bound with taxol. The drug-binding pocket of P-gp is globular in shape and utilizes forty residues from all TM helices to bind with taxol [21,65,66]. The binding pocket is characterized by a combination of aromatic residues, such as phenylalanine and tyrosine, and hydrophobic residues, including methionine and leucine (Figure 2A) [21,58]. In general, P-gp has a high affinity for globular hydrophobic, lipophilic, and weakly amphipathic molecules as the binding site is rich in aromatic and nonpolar residues such as phenylalanine and valine, leading to its characterization as a “hydrophobic vacuum cleaner” [15,18,19,20]. Aromatic residues facilitate the formation of biomolecular interactions such as cation–π and π–π interactions [67]. The delocalized electron cloud in the π bond may facilitate van der Waals bonds, allowing for greater flexibility in interactions with a diverse range of substrates [64,68]. TM4 and TM10, in particular, have been identified as critical components of the substrate binding site, forming a gate through bends in their structures. Interestingly, although mutations in either TM4 or TM10 alone do not significantly impact the transporter’s function, concurrent mutations in both TM helices disrupt its ability to transport drugs, suggesting that TM4 and TM10 play similar roles and the absence of one can be filled by the other [66].

X-ray crystallography has revealed a 6000 Å^3^ internal cavity in inward-facing P-gp, making it the largest such cavity identified in any protein [38,69]. The drug-binding cavity is characterized by a hydrophobic region, with only a small area being hydrophilic (Figure 2B) [21]. More recent studies have identified distinct structural features within P-gp where substrates and/or inhibitors bind, including a central drug-binding pocket, a vestibule, and an access tunnel (Figure 2A) [19,21]. The vestibule acts as an interface, linking the drug-binding pocket to the access tunnel. The access tunnel, located closer to the C-terminus of P-gp and surrounded by five TM helices, contains a cavity enriched in phenylalanine residues, which contributes to the transporter’s plasticity [22]. This tunnel extends from the central drug-binding site to the cytoplasmic gate, which is formed by TM4 and TM10 [19]. Inhibitors that bind beyond the drug-binding pocket, extending into the vestibule or access tunnel, restrict the movement of the TM helices, thus blocking the transport of ligands [23]. Additionally, this flexibility of the binding pocket could also have arisen due to the flexible linker region linking the two halves of P-gp [15,18,20]. Studies have shown that, even with 15 substitutions in the TMDs, P-gp can retain its drug transport function, underscoring its high molecular flexibility [70]. This flexibility allows major alterations in the drug-binding pocket to be well tolerated, maintaining the protein’s function.

## 4. Translocation Mechanism of P-gp

P-gp undergoes a dynamic cycle involving three main conformations: the inward-facing, occluded, and outward-facing conformation (Figure 3) [19,21,24,25,28,70]. These conformations are crucial for its function in transporting substrates, varying from 250 to 4000 Da, across the cell membrane against their concentration gradient [58].

Based on structural information obtained (Table 1), the translocation mechanism of P-gp has been proposed in several studies, and it can be a representative model for the role of the transporter in MDR (Figure 4) [21,23,24,27,28,36].

In the inward-facing conformation of P-gp, with the NBDs separated, transmembrane helices TM4 and TM10 are in a straight configuration, which can be interrupted by flexible loops that act as hinges, allowing the drug-translocation pathway to open [24,35,49]. The extracellular gate is closed while the drug-binding pocket located in the TMD is accessible from the interior of the cell, allowing binding of substrates within a pocket formed by residues from eight of the twelve TM helices [15,19,20,66].

Human P-gp has been determined in a ligand-bound occluded conformation in the presence of UIC2 or MRK16 Fabs [19,21,22,23,26]. Upon ligand binding in the central binding cavity, it is closed off from both sides, with TM4 and TM10 kinked, effectively sealing the gate from the cytosolic side [19,21]. The drug-binding pocket becomes sequestered, preventing access from both the cytoplasmic and extracellular environments. The occluded conformation is a transient intermediate in the cycle, transitioning rapidly to the next stage [71]. Due to its transient nature, this conformation is often challenging to detect; however, it has been characterized both in the presence of UIC2 and MRK16 Fabs to stabilize the intermediate state [53,72]. Competitive inhibitors are proposed to function by arresting the transporter in the occluded conformation to block substrate binding and conformational changes [21].

The transition to the outward-facing conformation requires ATP binding. Two ATP molecules bind to P-gp, with each ATP binding at the NBD dimerization interface interacting with distinct conserved motifs within the NBDs [72]. Furthermore, TM4 and TM5 in TMD1 and TM10 and TM11 in TMD2 from the two intersecting helices move inwards to move the NBDs closer together. The extracellular segment of TM7 and TM8 retract from TM9 to TM12, creating an outward-facing arrangement [24]. Interactions are formed between TM4 and TM6 and between TM5 and TM6, while bonds between TM1 and TM6 dissociate [30,58]. TM3 and TM6 also interact to contract the inner cavity and open the extracellular gate [30]. This outward-facing conformation facilitates the release of substrates into the extracellular environment. In this state, the substrate’s affinity for the drug-binding pocket decreases compared to the inward-facing conformation, allowing the substrate to be released into the extracellular environment [20]. Additionally, TM12, which forms a continuous α-helix along its full length, has also been found to push bound substrates out of the drug-binding pocket [19,73,74]. ATP hydrolysis then triggers a conformational change in P-gp, causing it to revert back to the inward-facing conformation, preparing P-gp for another cycle of substrate transport [18].

## 5. Structural Insights into P-gp Mutants and Their Role in Multidrug Resistance

The poly-specificity, or the ability of P-gp to accommodate multiple drugs, causes it to contribute significantly to MDR. By transporting various drugs out of cells, the intracellular concentrations and effectiveness of the drugs are reduced. Understanding these structural and functional dynamics is crucial for developing strategies to overcome MDR in cancer therapy.

### 5.1. Summary of Key P-gp Mutants Associated with Multidrug Resistance in Chemotherapy

To investigate the clinical impact of P-gp on patient outcomes, we conducted a comprehensive review of studies analyzing its mutations and their correlation with survival outcomes, such as overall survival (OS) and progression-free survival (PFS). These studies provided valuable insights into how mutational variation in P-gp affects the efficacy of chemotherapy in different cancer types. Additionally, we meticulously documented the specific drugs involved and the types of cancer being treated, aiming to draw more precise connections between P-gp mutations and survival outcomes (Appendix A). Consistently across multiple studies, T2677G/A, C1236T, and C3435T showed improved survival outcome.

### 5.2. Analysis of Specific P-gp Mutants Related to Cancer Treatment Outcomes

Six exonic mutations are related to cancer patients’ responses to drugs (Table 2 and Figure 5) [21,24]. Among these, the C1236T and C3435T mutations are silent, retaining their original amino acid, Gly, and Ile, respectively, despite a change in nucleotide sequence. The other four mutations are missense mutations that correspond to a change in amino acid. Specifically, rs9282564 corresponds to a nucleotide change of A61G in which Asn21 is mutated to Asp, which occurs in the cytoplasmic loop before TM1. rs2229109 corresponds to a nucleotide change of G1199A in which Ser400 is mutated to Asn, which occurs in the NBD1. rs2032582 corresponds to a nucleotide change of T2677G/A, in which Ser893 is mutated to Ala or Thr, which occurs in the cytoplasmic region of TM10. rs2229107 corresponds to a nucleotide change of T3421A in which Ser1077 is mutated to Thr in the NBD2 walker A motif.

### 5.3. Proposed Mechanisms of P-gp Mutants in Multidrug Resistance Based on Structural Insights

#### 5.3.1. A61G (Asn21Asp)

A61G occurs in the cytoplasmic loop before TM1. This mutation corresponds to a change in amino acid from Asn21 to Asp. Mixed results were obtained from the studies analyzing A61G P-gp. A study revealed similar expression levels and efflux activity between A61G P-gp and wild-type P-gp [75]. However, another study found that patients who are A61G homozygous require a lower dose of sertraline, a P-gp substrate, and experience fewer side effects [76]. Another study found that tacrolimus blood concentration was higher in patients with A61G P-gp during the first post-transplantation month [77].

Both Asn and Asp have a similar backbone; however, Asn is an amino acid with a polar, uncharged side chain, while Asp possesses a negatively charged side chain. The Asn21Asp substitution is located in the cytoplasmic loop region before TM1, a region with high flexibility. This makes it challenging for this region to be captured in structural studies. The N-terminal of P-gp may interact with NBDs during the translocation cycles, which may affect ATPase activity. The change in the amino acid may affect the folding of the protein and the dynamic feature of the flexible loop may indirectly modulate conformational changes, potentially influencing MDR mechanisms.

#### 5.3.2. G1199A (Ser400Asn)

G1199A occurs in NBD1, located before the walker A motif. The G1199A nucleotide change corresponds to a missense mutation where the original Ser400 amino acid is substituted with Asn. Past studies have revealed similar expression levels of P-gp in both wild-type and G1199A variants, while differences in activity in the G1199A variant were also detected [78]. G1199A requires a higher concentration of doxorubin to inhibit cell growth, indicating a higher degree of drug resistance compared to wild-type [79]. Furthermore, other studies also found the G1199A variant to be more resistant to vinblastine and vincristine [78]. G1199A P-gp has a greater ability to reduce absorptive transport across the epithelial barrier [80]. The study also found that, while V_max_ for the transport of ritonavir was similar in both G1199A P-gp and wild-type P-gp, the K_m_ value was two times greater in G1199A P-gp than for wild-type P-gp, indicating reduced transport affinity [80].

Both Ser and Asn are amino acids with a polar, uncharged side chain. Ser has a hydroxyl group which is a phosphorylation target, and it also participates in hydrogen bonding. On the other hand, the side chain of Asn, possessing an amid group, is slightly larger than Ser. The change of Ser400 to Asn may introduce differences in NBD1 which impact ATP binding or hydrolysis, thus affecting drug translocation cycles.

#### 5.3.3. T2677G/A (Ser893Ala/Thr)

T2677G/A occurs in the cytoplasmic region of TM10 [60,94]. The T2677G/A nucleotide change corresponds to a missense mutation where the codon TCT, encoding serine, is altered to GCT or ACT, leading to a potential Ala or Thr substitution at position 893. Ser893 is located far from the drug-binding pocket of P-gp and the NBDs. It is found in the cytoplasmic region of TM10 and located in the internal tunnel inner hinge domain between the NBD and has little effect on ATP binding and hydrolysis [96].

A study found that patients with T and A alleles, corresponding to Ser and Thr amino acid at position 893, are more responsive to FAC (Fluorouracil, Adriamycin, and Cytoxan) chemotherapy [81]. Furthermore, the G allele, corresponding to Ala at position 893, has also been found to have increased drug efflux activity, accounting for the reduced responsiveness to FAC chemotherapy [82]. As TM10 forms the lateral gate of P-gp and also plays crucial roles in forming kinks during substrate binding, a substitution of Ala at residue 893 of TM10 may affect the dynamic movement of TM10, thereby influencing the conformational changes of P-gp, resulting in higher resistance level during cancer chemotherapy.

#### 5.3.4. T3421A (Ser1077Thr)

T3421A occurs in NBD2. This mutation corresponds to a change in amino acid from Ser1077 to Thr. A study conducted found that T3421A P-gp is linked to higher plasma levels of phenytoin in individuals without disease when compared to those with wild-type P-gp [83]. This indicates that T3421A P-gp may have a higher activity level. Limited clinical studies were found studying the effects of T3421A mutation. As it locates at the walker A motif of NBD2, the mutation of Ser to Thr may affect the binding and stabilization of the ATP molecule at the binding site, altering the rate of drug efflux.

#### 5.3.5. C1236T (Gly412Gly)

The C1236T nucleotide change corresponds to a synonymous mutation where the codon GGC, encoding Gly, is altered to GGT, retaining the original Gly residue at position 412, located in NBD1. As the amino acid sequences are identical to the wild-type P-gp, there were no structural changes to C1236T P-gp. Yet, several studies have found C1236T to be associated with reduced MDR.

According to the codon statistics database, the Relative Synonymous Codon Usage (RSCU) of the original GGC codon is 1.346 RSCU while the mutated GGT codon is 0.646 RSCU [97,98]. The more than two-fold difference in RSCU values implies that the mutation from GGC to GGT may result in a less preferred codon being used to encode glycine in the protein. This could result in a slower translation rate, affecting the rate of formation of the tertiary structure of P-gp [99,100,101] and explaining why there is reduced efflux of drugs by P-gp and, hence, a reduced display of MDR.

Alternatively, the reduction in MDR could be explained by linkage disequilibrium. Several studies have found strong linkage disequilibrium between C1236T, C3435T, and T2677G/A [85,86,87]. This means that these genetic variants are often inherited together, and their combined effects could influence drug resistance. The reduced MDR observed in association with C1236T might, therefore, result from interactions with these other polymorphisms rather than the C1236T mutation alone. The potential interplay between these variants could alter the expression or function of P-gp in ways that contribute to the observed decrease in MDR.

#### 5.3.6. C3435T (Ile1145Ile)

The C3435T nucleotide change corresponds to a synonymous mutation where the codon ATC, encoding Ile, is altered to ATT, retaining the original Ile residue at position 1145 at NBD. As the amino acid sequences are identical to the wild-type P-gp, there were no structural changes to the C3435T P-gp. Yet, several studies have found the C3435T to be associated with reduced MDR and numerous functional studies have been conducted on the C3435T P-gp mutant.

Firstly, the literature has reported reduced P-gp activity in the C3435T mutant as compared to the wild-type [102]. Secondly, recent findings suggest that the C3435T P-gp mutant have decreased expression levels compared to wild-type genotype [88,90,92,93]. Similar to C1236T, this could be due to the codon usage bias, affecting expression level. According to the codon statistics database, the original ATC codon is 1.363 RSCU while the mutated ATT codon is 1.105 RSCU [98]. The mutation from ATC to ATT may result in a less efficient or less preferred codon being used to encode glycine in the protein [100]. A study has found that the homozygous T-allele is linked to P-gp expression levels that are more than twice as low compared to those in homozygous CC samples [90]. Furthermore, those with heterozygous genotypes show intermediate P-gp expression.

Fung and Gottesman (2009) also hypothesized that C3435T could impair the ribosome, altering the function and expression of P-gp [94,95]. Through interfering with the translation from 30 to 72 codons before the mutant site, the C3435T mutant could potentially alter co-translational folding. Examining the amino acid sequence suggests that an extra pause signal may disrupt the folding of two critical motifs, the Q-loop and the walker A motif, in NBDs [94].

## 6. Inhibitor Design and Potential Therapeutic Strategies

Inhibitors of P-gp have been extensively studied to enhance the efficacy of drug treatments to reduce MDR. Initial efforts with first-generation inhibitors, such as verapamil and cyclosporine A, encountered significant challenges due to their associated toxicity and adverse side effects [13,15]. These early inhibitors, while promising, were limited by their clinical safety profiles. To address these issues, researchers developed second-generation P-gp inhibitors, which were structurally designed to improve upon the shortcomings of the first generation. For example, valspodar, a second-generation inhibitor, demonstrated better effectiveness in reducing P-gp-mediated drug resistance. However, it also had its drawbacks, including the risk of causing the accumulation of certain cancer drugs to toxic levels, which limited its clinical utility. In response to these challenges, third-generation P-gp inhibitors were introduced, including zosuquidar, tariquidar, and elacridar (Figure 6) [19].

The three inhibitors, elacridar, tariquidar, and zosuquidar, function by binding in pairs to inhibit P-gp. The inhibitors mainly bind to the drug-binding pocket, the vestibule, and/or the access tunnel (Figure 2A and Figure 6) [19,103]. The access tunnel stretches from the central drug-binding pocket to the cytoplasmic gate formed by TM4 and TM10 [19]. The vestibule serves as the interface between the drug-binding pocket and access tunnel. The binding of the inhibitors to these regions restrict the movement of the TM helices, thereby inhibiting substrate transport [23].

For zosuquidar, one molecule binds to the drug-binding pocket while the other interacts with the vestibule. The binding area covers almost the entire width of the lipid membrane [22]. The two zosuquidar molecules enfold each other with a two-fold rotational symmetry, forming bonds with residues from eight TM helices (TM1, TM4, TM5, TM6, TM7, TM10, TM11, and TM12). Notably, four TM helices (TM4, TM6, TM10, and TM12) bend and kink to encircle the binding cavity, effectively blocking access to P-gp from the cytoplasmic region [22].

The binding mechanisms of elacridar and tariquidar differ slightly from that of zosuquidar. Like zosuquidar, one molecule of elacridar or tariquidar adopts a U-shaped, globular conformation that binds to the central drug-binding pocket. However, unlike zosuquidar, the other molecule adopts an L-shaped conformation, stretching beyond the vestibule to the access tunnel, where it functions as a noncompetitive inhibitor [19,26,103,104]. The access tunnel acts as a regulatory site in this context. Notably, prior research has shown that at low concentrations, elacridar and tariquidar can behave as substrates, when one molecule of elacridar or tariquidar adopts the globular conformation, and bind to the central drug-binding pocket [105]. Further, a recent study has also found three molecules of elacridar can simultaneously bind to and inhibit P-gp [23].

Common P-gp residues involved in interactions with inhibitors are presented in Appendix A. The majority of these residues are aromatic amino acids such as phenylalanine (F), tryptophan (W), and tyrosine (Y), providing multiple hydrophobic interactions with bound inhibitors.

These newer third-generation agents were designed with improved specificity and reduced toxicity. Despite their advancements, these third-generation inhibitors have not yet achieved broad regulatory approval for widespread clinical use. The ongoing challenges with P-gp inhibitors highlight the need for continued research to develop more effective and safer options. The focus remains on creating inhibitors that not only enhance drug efficacy but also minimize adverse effects and improve patient outcomes in the treatment of MDR-related conditions.

The findings discussed in Section 5 regarding the structural-functional aspects of P-gp mutants and the observed reduction in MDR in patients carrying certain mutants could be beneficial for precision therapy. Precision therapy is a form of medicine that involves using information about patients’ genes for treatment. Fully comprehending how different genotypes respond to different drugs and how they influence MDR could improve decision-making by clinicians [106].

Furthermore, designing a targeted inhibitor may also enhance treatment outcomes for these patients. Gene editing therapy has shown considerable promise. Specifically, modifying the gene encoding P-gp to genotypes associated with reduced MDR through gene-editing techniques like CRISPR/Cas9 has demonstrated potential in reducing MDR both in vivo and in vitro. For example, researchers have successfully utilized the CRISPR/Cas9 system to knock down the gene of P-gp, resulting in improved paclitaxel efficacy in colorectal cancer cells due to reduced drug efflux by P-gp [107]. Similarly, mutations introduced into sea urchin larvae targeting the NBDs of P-gp also led to reduced drug efflux [108]. Another study targeting exons 5 and 8 of P-gp achieved significant accumulation of rhodamine 123 and doxorubicin in MDR cancer cells by knocking out P-gp [109].

These studies collectively highlight the potential of gene editing to enhance drug efficacy by modulating P-gp activity. We propose that testing genetic variants T2677G/A, C1236T, and C3435T as targets could be particularly effective in addressing MDR. These variants could serve as strategic targets for gene editing, potentially minimizing off-target effects while still reducing P-gp activity to a level that enhances drug sensitivity without complete inhibition.

## 7. Areas for Further Research

The impact of genetic mutations on MDR in cancer cells has yielded conflicting results. For instance, the C3435T mutation has been linked to significant associations in both progression-free survival and overall survival in certain studies [6,110]. Conversely, other investigations have found no significant association [111,112]. This disparity in findings raises important questions about the role of genetic mutations in modulating MDR. Some research suggests that these mutations may enhance MDR by reducing the retention of chemotherapeutic agents within cancer cells, thus potentially leading to treatment failure [6,7]. In contrast, alternative studies have reported findings that challenge this notion, indicating that the same mutations might not consistently contribute to resistance [111,112]. As a result, the overall impact of genetic mutations on MDR remains unclear and requires further investigation. To address this uncertainty, we propose conducting a systematic review and meta-analysis to comprehensively evaluate the influence of these mutations on MDR. By aggregating data from multiple studies, a meta-analysis can enhance statistical power and identify trends or effects that may be overlooked in individual studies [113]. In addition, conducting experiments across various cell lines and in vivo models will provide a more robust framework for drawing definitive conclusions regarding the role of genetic mutations in MDR [114]. This multi-faceted approach aims to clarify the complex relationship between genetic mutations and drug resistance, ultimately contributing to more effective treatment strategies for cancer patients.

While inhibiting P-gp in the clinical setting seems to be a promising method to mitigate MDR, there remain no approved P-gp inhibitors for cancer treatment today [115]. This could be due to the limited understanding of P-gp in detail. Most studies are also limited in exploring the mechanisms through which P-gp mutations affect P-gp activity [114]. Cellular energetics and the ability of P-gp to respond to metabolic demands have also remained unexplored [115]. With studies often falling short of elucidating the underlying biochemical pathways involved, our understanding of how these SNPs influence P-gp activity at a molecular level is limited. Without a comprehensive exploration of these mechanisms, developing novel techniques to tackle MDR caused by P-gp poses a challenge.

Moreover, the effects of these mutations may vary based on the type of cancer and the specific drugs used. For example, while both vinblastine and verapamil are P-gp substrates, they have been found to have differing an affinity and permeability to P-gp [116]. Another study also compared six vastly different substrates of P-gp and found that they stimulate P-gp activity differently [117]. Therefore, future research could focus on further examining how these mutations impact MDR across different cancer types and with various chemotherapeutic agents. Such studies are crucial for clarifying the role of these genetic variants in MDR and for developing more effective, personalized treatment strategies.

In summary, while this review provides valuable insights based on structural information obtained, continued investigation into the role of mutations in different cancer contexts and with diverse drugs is essential to fully understand their impact on MDR.

## Figures and Tables

**Figure 1 ijms-26-04179-f001:**
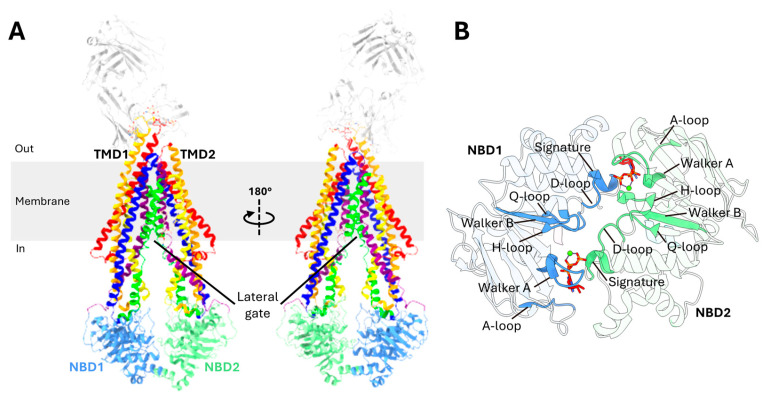
Overview of P-gp structure. (**A**) A 3D structural representation of P-gp (PDB code: 6FN4). The structure is shown in two transmembrane domains (TMD1 and TMD2), with the individual transmembrane helices color-coded for clarity. In TMD1, the helices TM1, TM2, TM3, TM4, TM5, and TM6 are colored red, orange, yellow, lime, blue, and purple, respectively. Similarly, in TMD2, the helices TM7, TM8, TM9, TM10, TM11, and TM12 are colored red, orange, yellow, lime, blue, and purple, respectively. The lateral gates are formed by paired TM4 and TM6, as well as TM10 and TM12. (**B**) View of the dimerized NBDs based on the outward-facing structure of human P-gp (PDB code: 6C0V). The ATP molecules are show as sticks and colored red.

**Figure 2 ijms-26-04179-f002:**
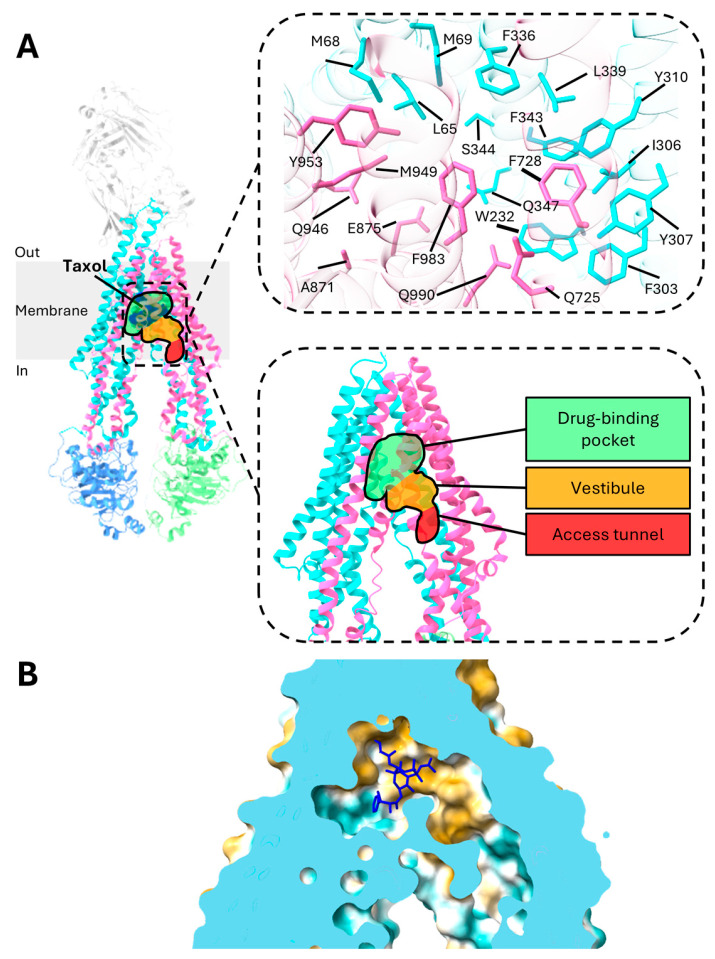
Substrate binding pocket of P-gp. (**A**) Structure of P-gp in complex with taxol (PDB code: 6QEX). The upper-right panel displays the zoomed-in view of the substrate binding pocket of P-gp, highlighting key interactions with taxol. To improve visibility, the taxol molecule itself is hidden, allowing for a clearer view of the P-gp atoms engaged in the interaction. The lower-right panel zooms in on the drug-binding pocket, vestibule, and access tunnel, which are highlighted in green, orange, and red, respectively. (**B**) Hydrophobic surface representation of P-gp (PDB code: 6QEX). The color gradient ranges from dark cyan (indicating the most hydrophilic regions) to dark goldenrod (representing the most hydrophobic areas).

**Figure 3 ijms-26-04179-f003:**
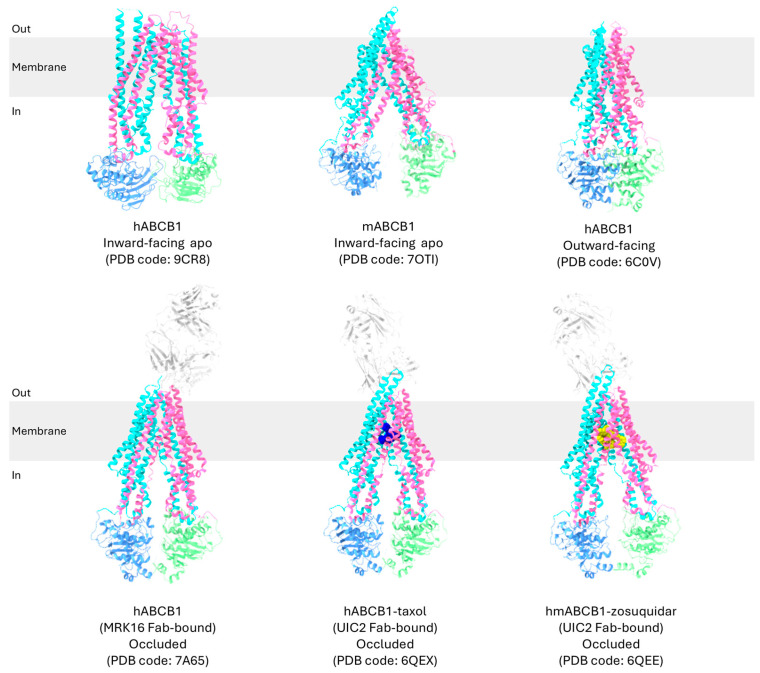
A 3D structural representation of P-gp in different conformations. TMD1, NBD1, TMD2, and NBD2 are colored cyan, blue, pink, and green, respectively.

**Figure 4 ijms-26-04179-f004:**
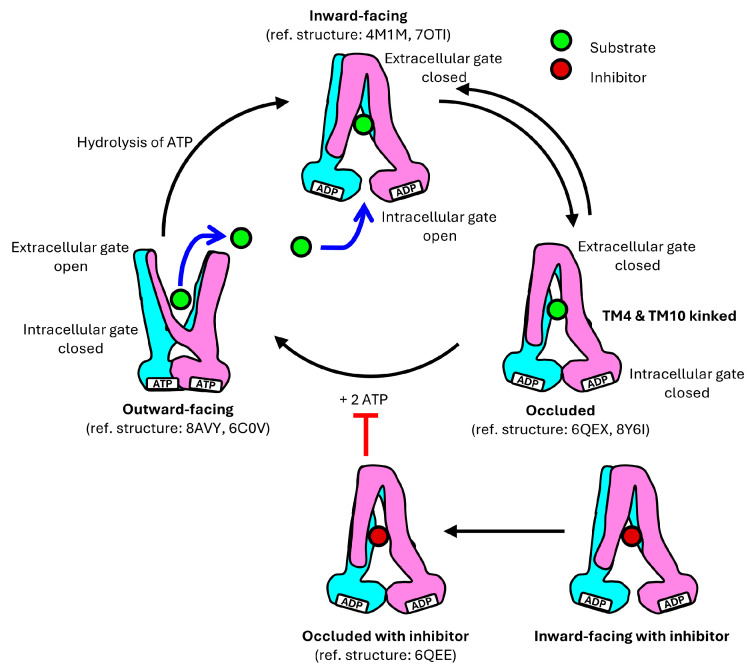
Proposed mechanism for the transport cycle of P-gp. In the inward-facing conformation, the NBDs are separated, allowing access to the drug-binding pocket from the interior of the cell, while the extracellular gate remains closed. In the occluded conformation, both the intracellular and extracellular gates are closed. In the outward-facing conformation, only the extracellular gate opens.

**Figure 5 ijms-26-04179-f005:**
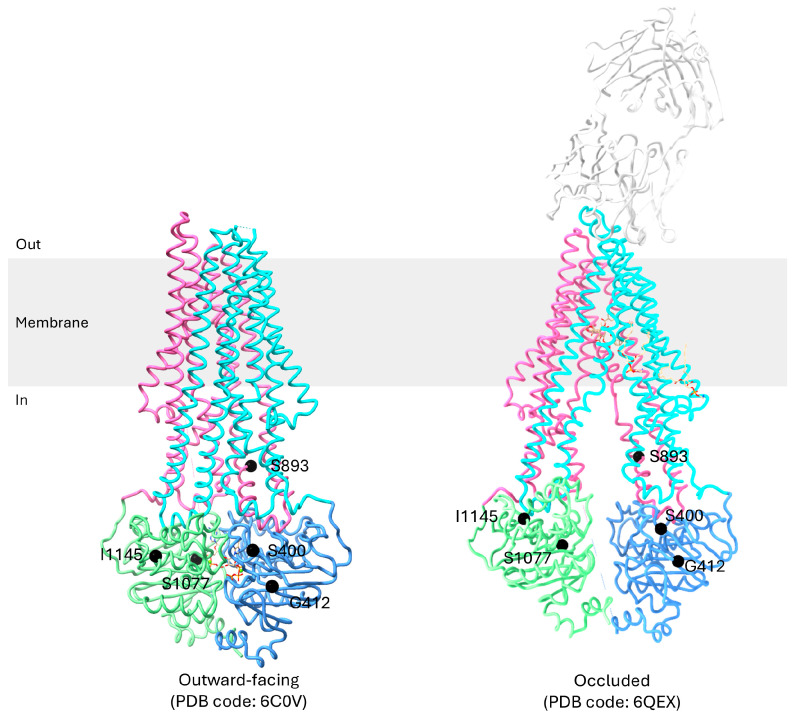
Structure overview of P-gp highlighting key mutations associated with multidrug resistance (MDR). The mutated amino acid residues are labelled and shown as black spheres. Note that Asn21 is not shown in the structure as it is not visible in the flexible N-terminal region.

**Figure 6 ijms-26-04179-f006:**
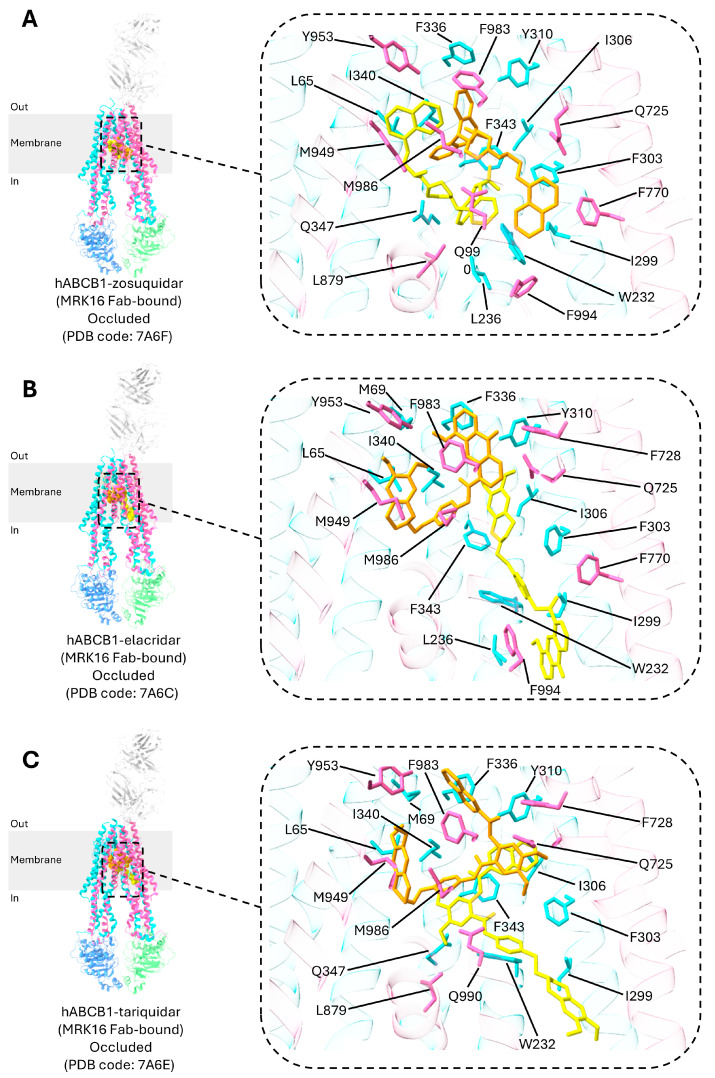
Structures of P-gp in complex with inhibitors. (**A**) Two zosuquidar molecules are enfolded around each other with a two-fold rotational symmetry. (**B**,**C**) The elacridar (**B**) or tariquidar (**C**) molecule adopts a U-shaped (orange), globular conformation that binds to the central drug-binding pocket, while the second molecule assumes an L-shaped conformation (yellow), extending beyond the vestibule and into the access tunnel. The inhibitors are represented as spheres in the main panel, with a zoomed-in inset on the right where the spheres are depicted as sticks, allowing for a clearer view of the binding interactions.

**Table 1 ijms-26-04179-t001:** Summary of structural data for P-glycoprotein (P-gp). The table lists PDB codes, the organism from which the protein was derived, the specific mutations studied, and the resolution of the structures. It also details the conformational state (inward-facing, outward-facing, or occluded), ligands bound, nucleotides involved, the method used, and the year of deposition. References to the original publications are provided for each structure.

PDB Code	Organism	Mutant	Resolution (Å)	Conformation	Ligand	Nucleotide	Method	Year	Ref.
9CR8	Homo sapiens		3.80	Inward-facing			cryo-EM	2025	[25]
9CTC	Homo sapiens		3.60	Occluded	Zosuquidar	ATP	cryo-EM	2025	[25]
9CTF	Homo sapiens		3.90	Inward-facing	Taxol	ATP	cryo-EM	2025	[25]
9CTG	Homo sapiens		3.40	Outward-facing		ATPγS	cryo-EM	2025	[25]
8Y6I	Homo sapiens, Mus musculus		2.54	Occluded	UIC2 Fab, elacridar		cryo-EM	2024	[23]
8Y6H	Homo sapiens, Mus musculus		2.49	Occluded	UIC2 Fab, elacridar		cryo-EM	2024	[23]
8PEE	Mus musculus	L335C	3.8	Inward-facing	AAC		cryo-EM	2024	[27]
8AVY	Mus musculus	L335C	2.9	Outward-facing		ATP	cryo-EM	2023	[27]
7ZKB	Mus musculus	V978C	4.7	Inward-facing	AAC		cryo-EM	2023	[27]
7ZKA	Mus musculus	V978C	2.9	Outward-facing	AAC	ATP	cryo-EM	2023	[27]
7ZK9	Mus musculus	L971C	4.3	Inward-facing	AAC		cryo-EM	2023	[27]
7ZK8	Mus musculus	L971C	3	Outward-facing	AAC	ATP	cryo-EM	2023	[27]
7ZK6	Mus musculus	L335C	3.1	Outward-facing	AAC	ATP	cryo-EM	2023	[27]
7ZK5	Mus musculus	L335C	2.6	Outward-facing	AAC	ATP	cryo-EM	2023	[27]
7ZK4	Mus musculus	L335C	2.6	Outward-facing		ATP	cryo-EM	2023	[27]
7O9W	Homo sapiens, Mus musculus		3.5	Occluded	UIC2 Fab, encequidar		cryo-EM	2022	[26]
7OTI	Mus musculus		4.2	Inward-facing			cryo-EM	2021	[28]
7OTG	Mus musculus		5.4	Inward-facing	Ivacaftor		cryo-EM	2021	[28]
7A69	Homo sapiens		3.2	Occluded	MRK16 Fab, vincristine		cryo-EM	2020	[19]
7A6F	Homo sapiens		3.5	Occluded	MRK16 Fab, zosuquidar		cryo-EM	2020	[19]
7A6E	Homo sapiens		3.6	Occluded	MRK16 Fab, tariquidar		cryo-EM	2020	[19]
7A6C	Homo sapiens		3.6	Occluded	MRK16 Fab, elacridar		cryo-EM	2020	[19]
7A65	Homo sapiens		3.9	Occluded	MRK16 Fab		cryo-EM	2020	[19]
6UJW	Mus musculus	Y306A, C952A	4.15	Inward-facing	BDE100		X-ray diffraction	2020	[29]
6UJT	Mus musculus	Y303A, C952A	4.17	Inward-facing	BDE100		X-ray diffraction	2020	[29]
6UJS	Mus musculus	F728A, C952A	4.17	Inward-facing	BDE100		X-ray diffraction	2020	[29]
6UJR	Mus musculus	F724A, C952A	4.1	Inward-facing	BDE100		X-ray diffraction	2020	[29]
6UJP	Mus musculus	F979A, C952A	3.98	Inward-facing	BDE100		X-ray diffraction	2020	[29]
6UJN	Mus musculus	C952A	3.98	Inward-facing	BDE100		X-ray diffraction	2020	[29]
6QEX	Homo sapiens, Mus musculus		3.6	Occluded	UIC2 Fab, taxol		cryo-EM	2019	[21]
6QEE	Homo sapiens, Mus musculus		3.9	Occluded	UIC2 Fab, zosuquidar		cryo-EM	2019	[21]
6A6M	Cyanidioschyzon merolae	Q147A/T381A	1.9	Outward-facing		AMP-PNP	X-ray diffraction	2019	[30]
6A6N	Cyanidioschyzon merolae	Q147A/T381A	3.02	Inward-facing			X-ray diffraction	2019	[30]
6Q81	Mus musculus		7.9	Inward-facing		ADP	cryo-EM	2018	[31]
6GDI	Mus musculus		7.9	Inward-facing			cryo-EM	2018	[31]
6FN4	Homo sapiens, Mus musculus	S559C, S1204C	4.14	Occluded	UIC2 Fab		cryo-EM	2018	[22]
6FN1	Homo sapiens, Mus musculus	S559C, S1204C	3.58	Occluded	UIC2 Fab, zosuquidar		cryo-EM	2018	[22]
6C0V	Homo sapiens	E556Q, E1201Q	3.4	Outward-facing		ATP	cryo-EM	2018	[24]
5KOY	Mus musculus	Δ649–682 (34 linker deleted)	3.85	Inward-facing		ATP	X-ray diffraction	2016	[32]
5KPJ	Mus musculus	Methylated	3.5	Inward-facing			X-ray diffraction	2016	[32]
5KPI	Mus musculus		4.01	Inward-facing			X-ray diffraction	2016	[32]
5KPD	Mus musculus	E552Q, E1197Q, 34 linker deleted	3.35	Inward-facing			X-ray diffraction	2016	[32]
5KO2	Mus musculus	E552Q, E1197Q, 34 linker deleted	3.3	Inward-facing	Hg^2+^		X-ray diffraction	2016	[32]
4XWK	Mus musculus		3.5	Inward-facing	BDE100		X-ray diffraction	2016	[33]
4Q9L	Mus musculus		3.8	Inward-facing	QZ-Phe		X-ray diffraction	2015	[34]
4Q9K	Mus musculus		3.8	Inward-facing	QZ-Leu		X-ray diffraction	2015	[34]
4Q9J	Mus musculus		3.6	Inward-facing	QZ-Val		X-ray diffraction	2015	[34]
4Q9I	Mus musculus		3.781	Inward-facing	QZ-Ala		X-ray diffraction	2015	[34]
4Q9H	Mus musculus		3.4	Inward-facing			X-ray diffraction	2015	[34]
3WMG	Cyanidioschyzon merolae	G277V, A278V, A279V	2.4	Inward-facing	aCAP		X-ray diffraction	2014	[35]
3WMF	Cyanidioschyzon merolae	G277V, A278V, A279V	2.6	Inward-facing			X-ray diffraction	2014	[35]
3WME	Cyanidioschyzon merolae		2.751	Inward-facing			X-ray diffraction	2014	[35]
4M2T	Mus musculus		4.35	Inward-facing	QZ59-SSS		X-ray diffraction	2013	[36]
4M2S	Mus musculus		4.4	Inward-facing	QZ59-RRR		X-ray diffraction	2013	[36]
4M1M	Mus musculus		3.8	Inward-facing			X-ray diffraction	2013	[36]
4KSD	Lama glama, Mus musculus		4.1	Inward-facing	NB592		X-ray diffraction	2013	[37]
4KSC	Mus musculus		4	Inward-facing			X-ray diffraction	2013	[37]
4KSB	Mus musculus		3.8001	Inward-facing			X-ray diffraction	2013	[37]
3G61	Mus musculus		4.35	Inward-facing	QZ59-SSS		X-ray diffraction	2009	[38]
3G60	Mus musculus		4.4	Inward-facing	QZ59-RRR		X-ray diffraction	2009	[38]
3G5U	Mus musculus		3.8	Inward-facing			X-ray diffraction	2009	[38]

**Table 2 ijms-26-04179-t002:** Summary of exonic P-gp mutations and their phenotypic effect and clinical implication.

SNP ID	Nucleotide Change	Amino Acid Change	Amino Acid	Mutation Type	Location	Phenotypic Effect and Clinical Implication
rs9282564	A61G	Asn → Asp	21	Missense	Cytoplasmic loop before TM1	- Similar expression level and efflux activity between A61G P-gp and wild-type P-gp [75]. - A61G homozygous patients require a lower dose of sertraline and have fewer side effects [76]. - Tacrolimus blood concentration was higher during the first post-transplantation month in patients with A61G P-gp [77].
rs2229109	G1199A	Ser → Asn	400	Missense	NBD1	- Similar expression level for wild-type and G1199A P-gp [78]. - Different activity level for wild-type and G1199A P-gp [78]. - G1199A P-gp more resistant to doxorubin, vinblastine, and vincristine than wild-type [78,79]. - Similar Vmax for the transport of ritonavir in both G1199A P-gp and wild-type P-gp while Km value was two times greater in G1199A P-gp [80].
rs2032582	T2677G/A	Ser → Ala/Thr	893	Missense	Cytoplasmic region of TM10	- T and A alleles more responsive to FAC chemotherapy compared to G alleles [81]. - G alleles have higher drug efflux activity [82].
rs2229107	T3421A	Ser → Thr	1077	Missense	NBD2 walker A motif	- T3421A P-gp has higher phenytoin plasma levels for individuals compared to wild-type P-gp [83].
rs1128503	C1236T	Gly → Gly	412	Silent	NBD1	- C1236T P-gp associated with longer progression-free survival and overall survival, suggesting reduced MDR [6]. - C1236T P-gp associated with lower P-gp expression and activity [84]. - Several studies have found strong linkage disequilibrium between C1236T, C3435T, and T2677G/A [85,86,87].
rs1045642	C3435T	Ile → Ile	1145	Silent	NBD2	- C3435T P-gp has reduced expression levels and efflux activity compared to wild-type [88,89,90,91,92,93]. - C3435 P-gp could have impaired ribosome binding [94,95].

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
