# Peer review of "Mapping the Role of P-gp in Multidrug Resistance: Insights from Recent Structural Studies"

_ijms, 2025, doi:10.3390/ijms26094179_

Round 1
Reviewer 1 Report
Comments and Suggestions for Authors
This review will provide valuable insights based on P-gp structural information obtained, continued investigation into the role of mutations in different cancer
contexts and with diverse drugs is essential to fully understand their impact on MDR.
Author Response
Thank you for your positive feedback. We truly appreciate your time and effort in evaluating our manuscript. Your comments are highly motivating and valuable to us to continue our research in this area.
Reviewer 2 Report
Comments and Suggestions for Authors
The article represents a clear, systematic overview of the structure, molecular mechanisms and clinical impact of mutations regarding P-gp and the implication in multidrug resistance related to various types of neoplasia. Figures and tables are clear and provide useful information on the structure, function and current knowledge of identified mutations and their impact on tratetment response of various chemotherapeutics. The authors also present extensively the various types of P-gp inhibitors as well as future research directions underlining the need for further studies to overcome P-gp-mediated resistance.
Author Response
Thank you for your positive feedback! We sincerely thank you for your time and effort in evaluating our manuscript. We truly appreciate your supportive comments and are encouraged by your assessment that our work is clear and well-structured.
Reviewer 3 Report
Comments and Suggestions for Authors
The transmembrane proteins carry out a central function in multidrug resistance. This review provides useful insights based on structural information to update the research. The role of mutations is elucidated in different cancer contexts for different drugs to fully understand their impact on MDR. This manuscript reviews the structural evidence-based mutational mechanisms involved in P-gp activity and discusses their impact on chemotherapy resistance. In addition, the review provides guidelines for P-gp inhibition, critical strategies for restoring drug sensitivity in resistant cancers, and outlines future research directions to combat P-gp-mediated MDR.
Reading the manuscript observed:
P1. L30. „New cancer cases are expected to surpass two million in 2024“ What does it mean by „expected“?
Tables 1 and 2 made it difficult to read the manuscript, and the information provided does not provide much useful information, so moving them to supplementary information would be a great solution.
P19. Table 3. Ser→Ala/Th r. The letter r is dropped.
P23. Table 4. I would recommend adding affinity or moving it to supplementary information, as the information is not essential.
No significant scientific deficiencies were observed in the manuscript.
Author Response
Comments 1: P1. L30. „New cancer cases are expected to surpass two million in 2024“ What does it mean by „expected“?
Response 1: Thank you for pointing this out. We agree that the statistic presented here is outdated and have updated to reflect the projected number of new cancer cases diagnosis in 2050. This can be found on P1 L29-L30.
Comments 2: Tables 1 and 2 made it difficult to read the manuscript, and the information provided does not provide much useful information, so moving them to supplementary information would be a great solution.
Response 2: Thank you for your valuable suggestion! We have moved Table 2 to supplementary information as Table S1. However, we think that Table 1 provides an overall summary of the determined structures and helps readers better understand the analysis in context. Therefore, we respectfully propose to keep Table 1 in the main manuscript.
Comments 3: P19. Table 3. Ser→Ala/Th r. The letter r is dropped.
Response 3: Corrected, thanks! Thank you for highlighting this. We have resized the table to fit the content.
Comments 4: P23. Table 4. I would recommend adding affinity or moving it to supplementary information, as the information is not essential.
Response 4: Thank you! We have moved Table 4 to supplementary information as Table S2.